# Addressing data management and analysis challenges in viral genomics: The Swiss HIV cohort study viral next generation sequencing database

Marius Zeeb[1,2*✪], Paul Frischknecht[1✪], Suraj Balakrishna[1,2], Lisa Jörimann[1,2], Jasmin Tschumi[1,2], Levente Zsichla[3,4], Sandra E. Chaudron[1,2], Bashkim Jaha[1], Kathrin Neumann[1], Christine Leemann[1], Michael Huber[2], Karoline Leuzinger[5], Huldrych F. Günthard[1,2✪], Karin J. Metzner[1,2✪], Roger D. Kouyos[1,2✪], The Zurich HIV Primary Infection Cohort Study, and the Swiss HIV Cohort Study

1 Department of Infectious Diseases and Hospital Epidemiology, University Hospital Zurich, Zurich, Switzerland, 2 Institute of Medical Virology, University of Zurich, Zurich, Switzerland, 3 Institute of Biology, ELTE Eötvös Loránd University, Budapest, Hungary, 4 National Laboratory for Health Security, ELTE Eötvös Loránd University, Budapest, Hungary, 5 Clinical Virology, University Hospital Basel, Basel, Switzerland

✪ These authors contributed equally to this work as first/last author.
* marius.zeeb@usz.ch

## Abstract

Numerous HIV related outcomes can be determined on the viral genome, for example, resistance associated mutations, population transmission dynamics, viral heritability traits, or time since infection. Viral sequences of people with HIV (PWH) are therefore essential for therapeutic and research purposes. While in the first three decades of the HIV pandemic viral genomes were mainly sequenced using Sanger sequencing, the last decade has seen a shift towards next-generation sequencing (NGS) as the preferred method. NGS can achieve near full length genome sequence coverage and simultaneously, it accurately encapsulates the within-host diversity by characterizing HIV subpopulations. NGS opens new avenues for HIV research, but it also presents challenges concerning data management and analysis. We therefore set up the Swiss HIV Cohort Study Viral NGS Database (SHCND) to address key issues in the handling of NGS data including high loads of raw- and processed NGS data, data storage solutions, downstream application of sophisticated bioinformatic tools, high-performance computing resources, and reproducibility. The database is nested within the Swiss HIV Cohort Study (SHCS) and the Zurich Primary HIV Infection Cohort Study (ZPHI), which together enrolled 21,876 PWH since 1988 and include a biobank dating back to the early nineties. Since its initiation in 2018, the SHCND accumulated NGS sequences (plasma and proviral origin) of 5,178 unique PWH. We here describe the design, set-up, and use of this NGS database. Overall, the SHCND has contributed to several research projects on HIV pathogenesis, treatment, drug resistance, and molecular epidemiology, and has thereby become a central part of HIV-genomics research in Switzerland.

**Data availability statement:** The use of the outputs generated by the database SHCND can be accessed in the framework of a collaboration with the SHCS and/or ZPHI, as the individual level datasets generated or analyzed during the current study do not fulfill the requirements for open data access: 1) The SHCS informed consent states that sharing data outside the SHCS network is only permitted for specific studies on HIV infection and its complications, and to researchers who have signed an agreement detailing the use of the data and biological samples; and 2) the data is too dense and comprehensive to preserve patient privacy in people with HIV. For collaborations a proposal should be send to the respective SHCS address (www.shcs.ch/contact). The provision of data will be considered by the Scientific Board of the SHCS and the study team. The full statement, from which the present statement is derived, can be found online: http://www.shcs.ch/294-open-data-statement-shcs.

**Funding:** This work was funded within the framework of the Swiss HIV Cohort study (SHCS) supported by the Swiss National Science Foundation (grant numbers 33CS30-201369 and 33FI-0 229621) (https://www.snf.ch/en) and by the Swiss HIV Cohort Study research foundation (https://shcsfoundation.ch/). The SHCS data are gathered by the Five Swiss University Hospitals, two Cantonal Hospitals, 15 affiliated hospitals and 36 private physicians (listed in in http://www.shcs.ch/180-health-care-providers). Furthermore, this work was supported by the Swiss National Science Foundation (grant number 179571 to H. F. G.); the Yvonne-Jacob Foundation (to H. F. G.) (https://stiftungen.stiftungschweiz.ch/organisation/stiftung-yvonne-jacob); the SHCS project 915 (to K.J.M.) and the University of Zurich Clinical Research Priority Program Viral Infectious Diseases, Zurich Primary HIV Infection Cohort Study (to H. F. G.) (https://www.viralinfectiousdiseases.uzh.ch/en.html). R. D. K. was supported by the Swiss National Science Foundation (grant numbers 324730_207957 and BSSGI0_155851). L.Z. was supported by the National Research, Development and Innovation Office in Hungary (RRF-2.3.1-21-2022-00006) (https://nkfih.gov.hu/about-the-office) as a part of the National Laboratory for Health Security. The funders had no role in study design, data collection and analysis, decision to publish, or preparation of the manuscript.

## Author summary

Medical data is becoming increasingly more complex, which significantly enhances research and clinical decision making. However, this growing complexity makes it also more difficult to handle it in a structured manner while adhering to good research practice and data management guidelines. In this context, we present the Swiss HIV Cohort Study Viral NGS Database (SHCND), a dedicated database storing and processing Next Generation Sequencing data (NGS) of HIV genomics data.

The SHCND centralizes all NGS data generated in the framework of the Swiss HIV Cohort Study (SHCS) and provides direct integration of bioinformatic pipelines for their processing. The SHCND streamlined the use of NGS data across researchers and was fundamental for a range of published research projects. Although developed to handle HIV NGS data, its flexible design makes it universally adaptable to any kind of data, for example, proteomics or imaging data. This work details the key design choices and functionalities of the SHCND aiming to serve as a practical guide for others seeking to establish databases for medical research data.

## Introduction

Viral genomic data plays a crucial role in HIV-1 medicine, research, epidemiology, and public health [1–25]. The global HIV-1 pandemic affects diverse populations, with differences in healthcare access and HIV-1 related health outcomes [26–29]. In this context, the high diversity of HIV-1 is of significant importance; notably to determine: drug resistance mutations (DRM) against anti-retroviral therapy (ART), viral transmission networks between people with HIV (PWH), viral pathogenesis, comorbidities linked to HIV-1, and immune responses, e.g., antibodies against HIV or auto immune diseases [10,30–33].

Until recently, Sanger sequencing was used to sequence at least the *pol* region (encoding for proteins relevant for viral replication) to determine the presence of DRMs [34–37]. However, over the last decade, next-generation sequencing (NGS), with technologies such as Illumina and Nanopore, increasingly replaced Sanger sequencing in both research and diagnostics. NGS allows a much higher throughput and easier sequencing of the entire viral genome [1,38]. NGS has several advantages over Sanger sequencing, especially the ability to detect (resistance) mutations present at low frequencies, i.e., when the mutation is only present in few percent of viral particles [39–41], and to determine within-patient diversity of HIV-1 genotypes as a marker of HIV-1 time since infection, transmission (if viral strains of two PWH are of high similarity), and super-infection (presence of two unique HIV-1 strains from independent transmissions) [31,42–44].

While NGS provides detailed information on the whole HIV-1 genome and offers benefits such as high-throughput and declining costs, additional challenges arise as it generates large data files and requires elaborate bioinformatic pipelines for interpretation [45–47]. Additionally, the lack of standardization for NGS data processing make it challenging to reproduce data reliably [48]: various NGS bioinformatic processing pipelines exist by different developers for applications such as genome assembly or DRM detection, each with different versions, input parameters [46] and often stochastic algorithms. Many tools are designed to produce the same output in principle, such as a genome alignment, but each tool-/version-/parameter choice can subtly influence the actual outcome leading to results to differing between tools.

**Competing interests:** I have read the journal's policy and the authors of this manuscript have the following competing interests: Within the last 5 years, K.J.M. has received travel grants and advisory board honoraria from Gilead Sciences and ViiV; and the University of Zurich received research grants from Gilead Sciences and Novartis for studies that Dr Metzner serves as principal investigator. H. F. G. has received research grants from the Swiss National Science Foundation, Swiss HIV Cohort Study, Yvonne Jacob Foundation, NIH, Gilead, ViiV, and is a subcontractor to a Bill and Melinda Gates foundation grant, paid to his institution; personal honoraria for data safety monitoring board or advisory board consultations from Merck, ViiV healthcare, Gilead Sciences, Janssen, Johnson and Johnson, Novartis, and GSK; and personal travel expenses from Gilead. All other authors declare no conflicts of interest.

Considering these obstacles, detailed documentation of all processing steps is crucial for the reproducibility and comparability of results in case some steps need to be redone or altered. This needs special consideration, even more so, because software complexity will only increase further, with trends to specialization on narrow tasks. Therefore, the research protocol documentation for each sample should contain all corresponding analyses in detail, including the version and specific running parameters of the tool used. If this is fulfilled, output data storage is not even required, as results can be reproduced using the documentation. One potential exception are non-deterministic analyses (processes which have different outcomes despite the same input due to randomness in the algorithms), for example, the sub-sampling of NGS reads. While modern software mostly allows specifying a so called random "seed" to reproduce the same (pseudo-)random results, some software might be hard-coded to determine a "seed" based on the current time or hardware random number generators. Finally, if external (unversioned) databases or web services outside of the project's control are utilized, their results need to be stored as they might not be reproducible later. In consequence, such data storage and computational solutions are required, which satisfy the conditions of sensitive human health data and have the ability to integrate raw data, processed data, and executable copies of the bioinformatic pipelines used.

A modern, future proof NGS database must meet all these demands. We, accordingly, developed therefore, the Swiss HIV Cohort Study Viral NGS Database (SHCND), in the framework of the Swiss HIV Cohort Study (SHCS) and Zurich Primary HIV Infection Cohort Study (ZPHI) [49,50] to ensure the reproducible use of viral NGS data. This work aims to describe the design, set-up, maintenance, and use of the SHCND, both from a developer- and a user-/researcher perspective.

## Methods

### Aim

The SHCND aims to provide a centralized storage and compute-orchestration solution over the whole digital life cycle of a HIV-1 NGS record, i.e., from raw NGS read data to all relevant derived processing outputs. For the purpose of reproducibility, all processed outputs are linked to the exact version of the originating pipeline with all its dependencies. Redundant processing steps, like NGS sequence assembly, are streamlined and executed only once, rather than separately for each analysis. The assembly is then accessible to all researchers, saving both computational resources and time. Currently the SHCND handles mostly NGS data from Illumina currently. However, the SHCND is designed with a modular and extensible design, allowing for the convenient incorporation of new bioinformatic pipelines and data, particularly when they are available in a containerized from.

### FAIR guiding principles

The SHCND follows the FAIR (Findability, Accessibility, Interoperability, and Reusability) Guiding Principles for scientific data management [51]. In brief, it makes data Findable and Accessible, by assigning globally unique and persistent (immutable) identifiers (F1, A1) and by use of standardized HTTPS communication protocols to handle access and authentication (A1.1, A1.2). It uses standard JSON format for rich and extensible metadata capture (F2, F3, A2), which is transformed to an indexed and searchable graph database representation (F4). The system is Interoperable with freely available analysis and processing solutions, specifically targeted at NGS data processing (I1, I3). The database is Reusable with management of data provenance and reproducibility of results, and we argue that reuse of our data and metadata is limited only by medical data privacy (R1, R1.1, R1.2). We also make the case that

the infrastructure and design of the system are reusable for other similar efforts and give an outlook to some of the many possible extensions, uses and existing reuses of the system.

## Sampling & NGS workflow

In the basic workflow from sampling to digital HIV-1 sequence described in Fig 1, a blood sample (plasma for viral RNA, PBMC for proviral DNA) is collected from PWH and then processed either for storage in the SHCS/ZPHI biobank or for direct analysis. The SHCND includes data from samples requested for various research projects from centers all over Switzerland, which are shipped for sequencing primarily to the SHCS Laboratory in Zurich and in some cases internationally, for example, to the Sanger Institute in Oxford [8]. NGS of near full-length HIV-1 sequences is performed either from plasma RNA or from proviral DNA. The preparation, amplification, and sequencing protocols are described elsewhere in detail for plasma RNA [42,52] and proviral DNA [53,54].

## SHCND workflow

The steps in the workflow described in Fig 1 were significantly centralized and streamlined with the introduction of the SHCND – prior to its development, similar steps were carried out by each researcher individually with NGS raw reads files stored on a networked file share. The

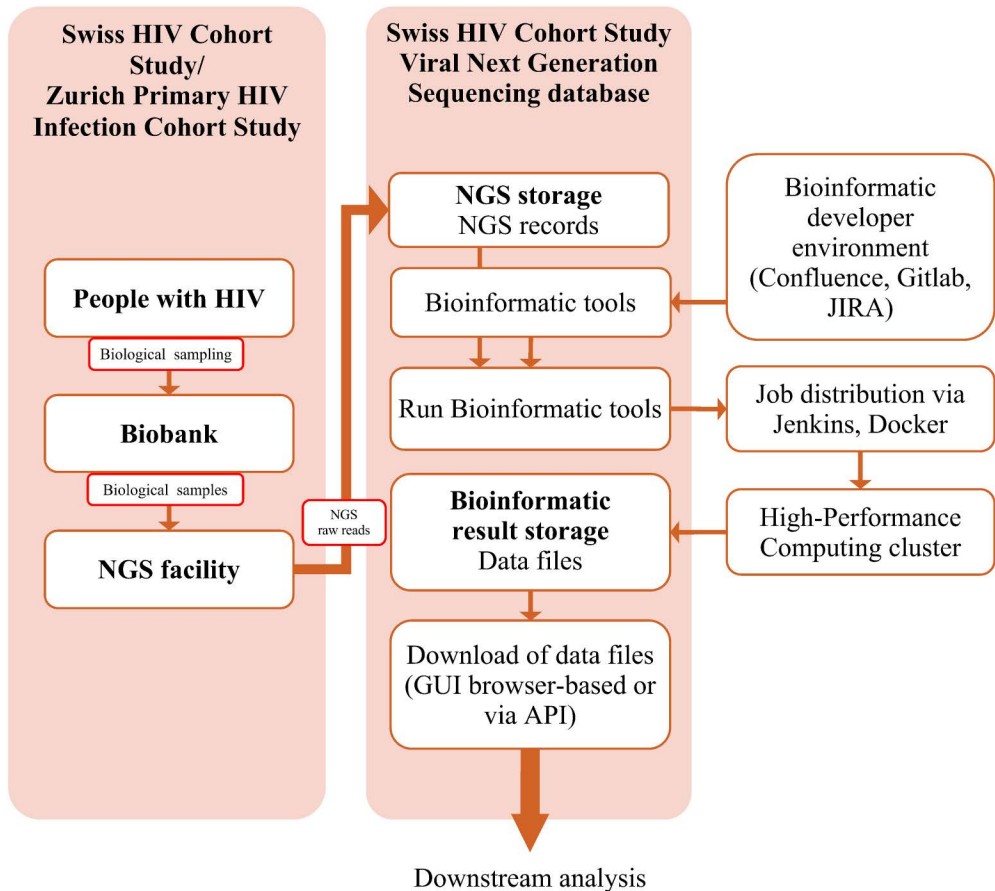

**Fig 1. From sample to genome.** Description of the steps from blood sampling of people with HIV, Next Generation Sequencing and its data handling, to the bio-informatics tools for computational HIV genome analysis.

NGS records (NGS raw reads in the form of fastq files) are uploaded to the SHCND together with the respective metadata (e.g., sample date, primer information), excluding sensitive patient data. Each uploaded NGS record is assigned a 36-character alphanumeric randomly generated universally unique identifier (UUID version 4), such as "5bfc99f6-8432-4afc-be32-3f9d2dfa4871" (Fig 2A).

The use of UUID version 4 identifiers [55] ensures globally unique, unequivocal, context-free, immutable identification that can be auto-generated in a distributed fashion as opposed to e.g. sequential integer IDs or human readable names, free from any privacy or ordering concerns. UUIDs are in widespread use in technical and non-technical domains for physical and digital artifact identification (e.g., [56–58]). Their special notation makes them particularly easy to recognize and parse as identifiers and we use this extensively to allow users to copy-paste lists of such UUIDs that can be separated by any or no characters at all.

All generated data files on the database receive their own UUID as well and are linked to the corresponding NGS record and processing tool through additional metadata files (Fig 2B). From the SHCND, any number of NGS records can be chosen by their UUID and submitted for parallel processing with a selected well-defined version of a bioinformatic tool. Once the user confirms the selection of NGS records, tool, and relevant parameter settings (appropriate for the NGS record, e.g., NGS platform and amplification method), the data files are transferred to an external high-performance-computing cluster, where they are processed. The results are transferred back to the SHCND where they are stored and linked with the sample by their UUIDs. For the retrieval of results, the user must specify again the UUIDs of the NGS records and the relevant tool to initiate a download. More sophisticated means of querying

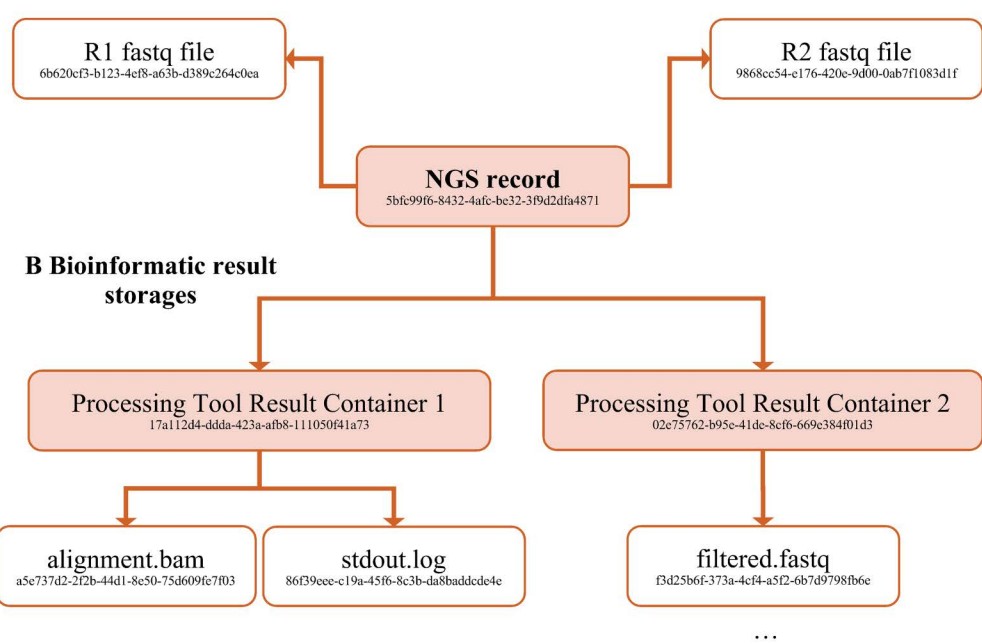

**A NGS record structure**

R1 fastq file
6b620cf3-b123-4ef8-a63b-d389c264c0ea

R2 fastq file
9868cc54-e176-420e-9d00-0ab7f1083d1f

NGS record
5bfc99f6-8432-4afc-be32-3f9d2dfa4871

**B Bioinformatic result storages**

Processing Tool Result Container 1
17a112d4-ddda-423a-afb8-111050f41a73

Processing Tool Result Container 2
02e75762-b95e-41de-8cf6-669e384f01d3

alignment.bam
a5e737d2-2f2b-44d1-8e50-75d609fe7f03

stdout.log
86f39eee-c19a-45f6-8c3b-da8baddcde4e

filtered.fastq
f3d25b6f-373a-4cf4-a5f2-6b7d9798fb6e

…   …

**Fig 2. Illustration of the database storage.** (A) NGS storage and (B) Bioinformatic result storage implementation in SHCND. Basically, a graph of UUID labeled nodes is formed. The graph structure (edges) itself is also materialized in UUID-named files containing JSON (not shown). Furthermore, the exact processing tool used to generate each result container is included in the metadata of each result container.

for results obtained with specific tool versions and parameters are possible, as the SHCND persistently stores the entire metadata of each parametrized run of a processing tool.

## SHCND IT architecture

On the highest level, the SHCND consists of two types of subsystems: firstly, the singular database core, which is responsible for data storage, and secondly, a set of client nodes, which interact with the core to request and handle computations. Client nodes interact with the database in three operating modes: performing queries (read), uploading data (write), and performing computations (read-write). These client nodes are typically either computers of researchers or virtual machines running in a self-service private cloud operated by the University of Zurich scientific IT services (an OpenStack instance) [59]. Client nodes controlled by human operators can perform manual (bulk) interactions through the SHCND web UI. In particular they can request NGS processing computations and query and download data. For data uploads, data querying, and bulk downloads, a Hypertext Transfer Protocol Secure (HTTPS) application programming interface (API) is used in dedicated client applications or scripts. The database core hosts the web UI and API.

The automatically operated virtual machine clients handle any outstanding requested computations. Initially, outstanding computations were part of the database system state and required custom implementation of coordination strategies to ensure they would run non-redundantly in parallel. However, we have since migrated computational job assignment to worker nodes to a private instance of Jenkins [60], a widely used open source "automation server" tool. In brief, Jenkins stores a list of pending tasks and schedules them to run on a set of worker nodes, known as agents. It evenly distributes a specified maximum number of parallel jobs across the worker nodes. We chose Jenkins for its web UI, which provides real time logs of running or past processing tools and facilitates easy observation of errors in individual jobs. We are also considering using a SLURM-managed compute cluster as an additional or alternative job submission system implementations [61].

## Bioinformatic pipelines

The bioinformatics tools or pipelines (detailed below) are executed as Docker containers [62] on Jenkins agents (Fig 3). We maintain a git repository with a Dockerfile for each tool. Each tool is built once for a given version with a fixed build time and fixed dependencies (potentially obtained from external sources), resulting in an immutable Docker container image. This image is used for all processing jobs requiring the respective tool with the specified version. In a nutshell, Docker container images can be understood as virtual machine images or a complete (Linux) file system tree. Processes running inside a Docker container are isolated from the host machine except for selective access to the file system and network interface. Another tool for the same job that we consider using is Singularity [63]. Before tool execution, we copy the necessary input files, such as fastq files, from the SHCND into the container and rename them to adhere to the expected file names of the respective tool. After successful processing, we collect result files from a predetermined folder before deleting the container on the agent. The result files are uploaded to the SHCND and incorporated into the file system as a set of result files and new metadata files, which link the NGS record to the new processing result (Figs 2/3).

While the git repositories containing the Dockerfiles and the Docker container images built from them are not directly stored as immutable data files in the SHCND (unlike the input and output data files of all computations), they are nevertheless an integral part of the reproducibility strategy of the system. Every version of every tool and data file is persistent, immutable

and remains accessible. Any "modifications" to data and computation tools are represented as new versions in an append-only version history across the system.

In addition to running the tools directly on the raw NGS record files of an individual sample, they can also be applied to any combination of previous processing tool output files associated with that individual sample. Furthermore, both raw NGS files and output files from a whole set of samples can be processed together with what we call "global" processing tools (as opposed to the usually local or per-sample processing tools used on a single sample at a

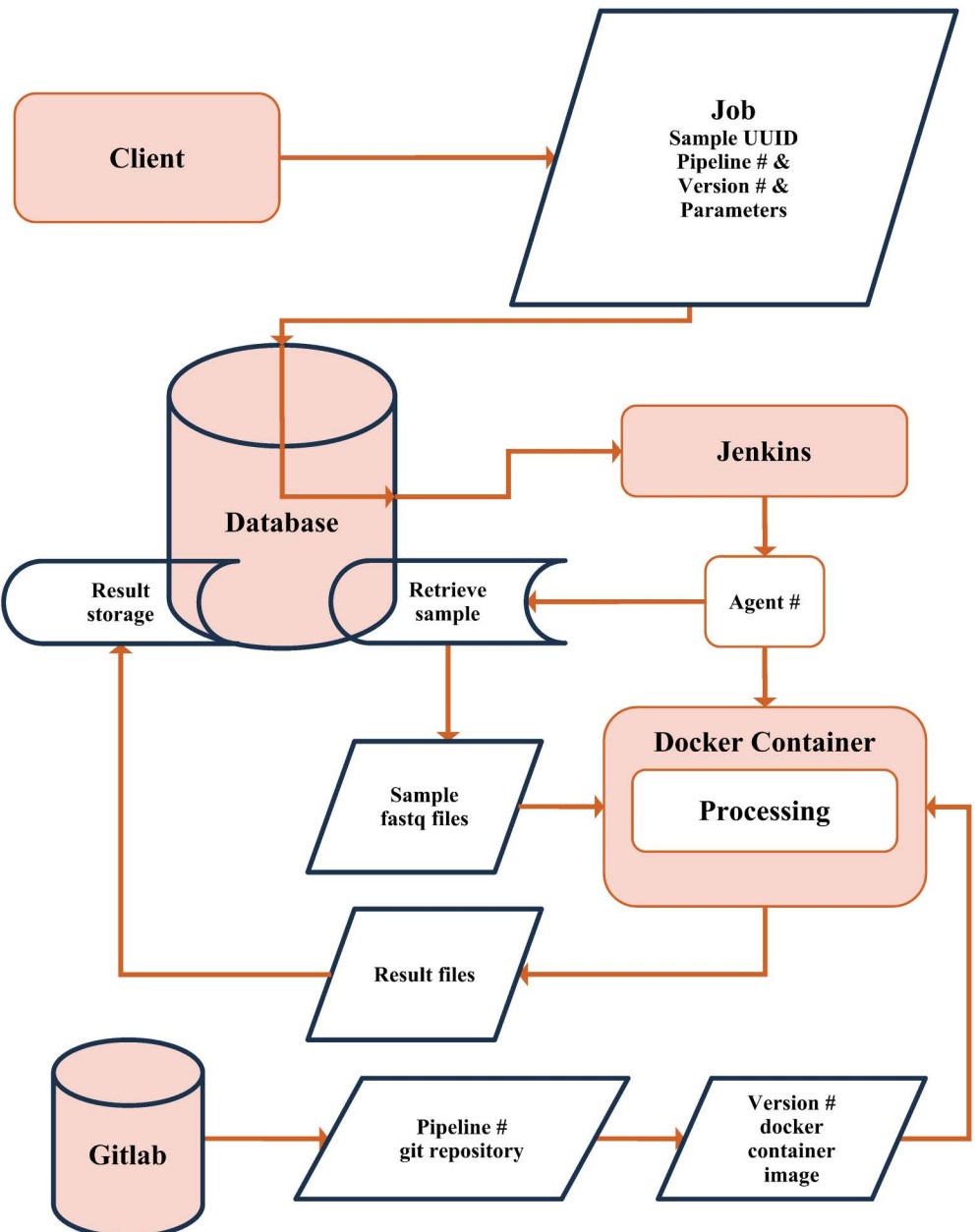

**Fig 3. Illustration of the bioinformatic pipeline processing.** From initial job submission to storage of processed results. In the case of "global" processing tools, multiple samples' fastq and output files from previous processing tool runs can be retrieved as inputs to the Processing step (not shown).

time). This capability allows for the use of tools like blast or the construction of phylogenetic trees on any subset of samples from the database, as discussed below.

## File management

The database core stores and manages all data as immutable flat binary files with an assigned UUID. It also coordinates all client node interactions, serving both the web UI and HTTPS API. An initial priority was, that a combination of large unstructured (from the SHCND point of view) data files, semi-interpreted metadata, and evolvable structures of links between this data must be stored. This requirement made traditional SQL/relation table-based database structures unsuitable, as they would require significant structural adjustments during the evolution of the system. For reasons of reproducibility, any and all changes to the data had to be fully versioned and in principle undoable. Furthermore, by the original design for job distributions and other features, it was a requirement that database status changes, e.g., pending job requests, must be communicated to connected clients to inform their next processing decisions or update their redundant/denormalized data replicas or projections of database state. We wanted a design where meta- and linkage data could be converted in real-time into a structured, queriable format (a "projected" data replica) as soon as it became available in the SHCND. During prototyping, metadata was transformed and submitted to a (read only) PostgreSQL database for querying needs as soon as it became available. Currently, we maintain a non-rigid structured replica in the form of a neo4j graph database [64]. Importantly, such ephemeral projected read models or read replica representations of the data in the database can be evolved extremely rapidly and flexibly to fit any querying needs without the need to physically restructure the historic versioned master information preserved in the database core.

For these reasons, the central SHCND storage was conceptually designed as a single, append-only, fully sequential list or "log" of UUID-named immutable binary files, also known as binary large objects (blobs). This is sometimes called an event-driven- and more precisely an Event Sourcing software architecture [65]. Any permanent information added to the SHCND is appended to this log/list of "events" as a blob. The only atomic operation required of the system is the assignment of successive sequence numbers to each new blob. The event of adding a blob is immediately communicated in-order to all connected clients via a WebSocket connection [66]. In this way, the blob log takes on the additional role of a shared message bus or event queue.

A blob is added, for example, to represent the initial metadata state of a new NGS record as an empty JSON object. The server assigns the next sequence number and a UUID (e.g., "5bfc99f6-8432-4afc-be32-3f9d2dfa4871") as the identifier for this new blob. Conceptually, we also use this same UUID as what we call a "base UUID" to refer to the entire version history of this NGS record, and in particular to refer to its latest revision at any point in time. To illustrate how revisions are handled, let us continue the example: an additional blob can be added to mark any sample changes, for example, metadata corrections or file attachments (the case of file attachment is illustrated in Fig 2A and Table 1). The new blob is assigned a new random UUID (e.g., "ed7ed2c3-ea61-4e33-86da-49c3eea347d8") and contains an updated JSON representation of the NGS record file and its metadata. A third blob, whose content is marked with a special Update-UUID-Prefix-Constant, then communicates that "ed7ed2c3-ea61-4e33-86da-49c3eea347d8" is an updated version of "5bfc99f6-8432-4afc-be32-3f9d2dfa4871". A dedicated client process monitors these update messages and maintains queriable derived databases of the "latest revision" and "version history" of all blobs. These derived databases thus are other examples of projected read models of the database. We note that most blobs containing data files are never updated in this fashion.

**Table 1. Simplified illustration of how the NGS record with base UUID 5bfc99f6-8432-4afc-be32-3f9d2dfa4871 might have been created in the database core's blob log as a series of UUID-named files whose content is interpreted to yield the SHCND subgraph shown in Fig 2A. The sequence number arbitrarily starts at 1000 and there are no other blobs created between the initial creation of the NGS record and the uploading and attachment of its fastq files.**

| SEQUENCE# | Blob uuid | Content | Meaning of content |
|---|---|---|---|
| 1000 | 5bfc99f6-8432-4afc-be32-3f9d2dfa4871 | {} | Empty JSON object. Base revision of an NGS record. |
| 1001 | 6b620cf3-b123-4ef8-a63b-d389c264c0ea | @M02081:266:000000000-BCBNY:1:1101:17805:18111:N:0:3<br>GGTTCTATAAAACTCTGAGGGCCGAGCAAG…<br>+<br>BBA… | R1 fastq file content. |
| 1002 | 479e8058-532a-49bc-8e9e-ca542f132561 | @M02081:266:000000000-BCBNY:1:1101:17805:18112:N:0:3… | R2 fastq file content |
| 1003 | ed7ed2c3-ea61-4e33-86da-49c3eea347d8 | {"R1": "6b620cf3-b123-4ef8-a63b-d389c264c0ea", "R2": "479e8058-532a-49bc-8e9e-ca542f132561"} | New revision content for the NGS record, now linking it to the two fastq files |
| 1004 | b8ec2f65-e195-44c4-a28e-9b551056decd | <UPDATE-UUID-PREFIX-CONSTANT>,5bfc99f6-8432-4afc-be32-3f9d2dfa4871,ed7ed2c3-ea61-4e33-86da-49c3ee-a347d8 | Tells the system that "ed7ed2c3-ea61-4e33-86da-49c3eea347d8" is an updated version of "5bfc99f6-8432-4afc-be32-3f9d2dfa4871" |

Originally, the blob-append mechanism was also used to implement a remote-procedure call (RPC) mechanism to facilitate command/query-response protocols between clients. A first specially tagged blob would communicate a service request to another client process implementing the requested functionality. The responding client would observe the request, process it, then post a suitably tagged response to the blob log. However, this approach had a drawback: read-only operations, such as querying the latest blob revisions, unnecessarily extended the blob log with ephemeral, computable information. Therefore, read-only queries are now implemented directly as ephemeral HTTP API calls. Only persistent state changes (commands) are durably stored in the log. The special revision marker or "update" blob mentioned above can be seen as an example of such a durably stored command, though the client process handling it doesn't post any response to the log in reaction to it.

The monolithic core of the system is written in isomorphic TypeScript (migrated from JavaScript), running in both nodejs (server-side) and web browser (client-side) JavaScript engines [67,68]. Other technology choices include Docker for containerization, Jenkins for job orchestration, and MongoDB for fast blob metadata querying [60,62,69]. All server-side system components run in Docker containers based on Ubuntu Linux, while the client-side web UI works in any modern browser and consists of basic HTML layouts using jQuery for selected interactivity [70]. As for deployment, apart from the web UI that can instantly be invoked from any authorized web-browser, dedicated scripts and client programs are downloaded manually to client computers where bulk downloads or uploads of data shall be performed. The database core and Jenkins agent virtual machines are set up manually. GitLab CI (continuous integration) scripts are used to deploy any enhancements to the database core virtual machine automatically on each git commit. There is no obstacle in principle to further automate the few manual steps, should the need arise.

## SHCND scientific pipelines

Currently implemented in the production environment of the SHCND and predominantly used are the following bioinformatic pipelines:

1. SmaltAlign

SmaltAlign performs consensus genome assembly using a reference sequence and an NGS record file (a fastq file). In brief, the pipeline performs an initial *de novo* assembly-assisted alignment against a chosen HIV-1 reference (e.g., HXB2, GenBank accession number K03455 [71]) followed by iterative alignments against the newly generated consensus. The alignment is output as a BAM file. The retrieved consensus sequences exhibit high quality, comparable to other commonly used assembly pipelines like shiver or V-pipe [72–74]. Detailed descriptions of the design and parameter settings can be found on https://github.com/medvir/SmaltAlign.

2. MinVar

MinVar uses the NGS record file (a fastq file) to detect drug resistance mutations (DRMs). In brief, MinVar samples sequencing reads to determine the HIV-1 subtype, followed by generating a consensus sequence from which it ascertains mutations, and compares those to known DRMs maintained by the Stanford drug resistance database [75]. This pipeline is also used in diagnostic settings and was validated accordingly [30]. Detailed descriptions can be found on https://ozagordi.github.io/MinVar/.

3. Hypermutation read filter

Hypermutation read filter uses the BAM file, i.e., mapped reads aligned to HIV-1 HXB2, generated by SmaltAlign to determine if reads are hypermutated according to Hypermut 2.0 [76]. The output is a fastq file with hypermutated reads filtered out, which can then be used as clean input for subsequent runs of SmaltAlign and MinVar instead of the unfiltered NGS raw reads file of the sample.

## Results

### Swiss HIV cohort study viral NGS database sequence collection

The SHCND contains 8,015 NGS records from 5,178 (24%) out of the 21,876 PWH ever enrolled in the SHCS or ZPHI (Table 2). While the SHCND covers all major demographic groups, transmission modes, and HIV-1 subtypes, several demographic groups among the SHCS participants are currently overrepresented, due to the fact that NGS was performed in the framework of specific projects [8,43,53,54,77–80]: PWH with any sequence (proviral DNA or plasma RNA) in the SHCS are on average more often white (76.7% NGS vs 64.6% no NGS) leading to an underrepresentation of ethnical minorities (3.4% NGS vs 18.9% no NGS), have HIV-1 subtype B (64.4% NGS vs 47.1% no NGS) leading to an underrepresentation of rare subtypes (16.6% NGS vs 38% no NGS), and were enrolled into the SHCS more recently (median calendar year 2005 NGS vs 1997 no NGS). Additionally, they are more likely to have acquired HIV via homosexual contact (45.2% NGS vs 37.5% no NGS) and less likely through intravenous drug use (10.5% NGS vs 17.1% no NGS) (Table 2).

Gene sequence availability (Fig 4A) from plasma RNA samples ranges from 3,567 *vif* sequences up to 3,678 *gag* sequences. For sequences derived from proviral DNA, it ranges from 3,264 *env* sequences up to 3,806 *nef* sequences. 3,287/3,795 (87%) sequences from plasma RNA cover all nine genes, compared to 2,829/4,220 (67%) sequences from proviral DNA. The earliest sample was already collected in 1988, whereas NGS started only in 2013 (Fig 4B).

Several original research articles harnessing the SHCND have already been published in international peer-reviewed journals (Table 3). A precursor database was used to infer the impact of the viral genome on the size and heritability of the HIV-1 reservoir and the prevalence of HIV-1 proviral drug resistances [54,78,83]. Several projects then used the SHCND to study antiretroviral resistance. A comparison between Sanger sequencing and NGS for DRM detection showed comparable performance, but NGS detected more low frequency DRMs

**Table 2. Characteristics of people with HIV stratified by HIV NGS availability and cohort.**

| | Overall | SHCS[1] | | ZPHI[2] | |
|---|---|---|---|---|---|
| | | no NGS | NGS | no NGS | NGS |
| N (PEOPLE) | 21,876 | 16,606 | 4,759 | 92 | 419 |
| FEMALE SEX (%) | 5,891 (26.9) | 4,583 (27.6) | 1,280 (26.9) | 8 (8.7) | 20 (4.8) |
| YEAR OF ENROLLMENT (MEDIAN [IQR]) | 2000[1992,2010] | 1997[1991,2009] | 2005 [1998,2010] | 2016 [2008,2020] | 2009[2006,2014] |
| YEAR OF BIRTH (MEDIAN [IQR]) | 1964[1958,1972] | 1963[1957,1971] | 1966[1959, 1974] | 1978[1967, 1986] | 1974[1967,1981] |
| LIKELY HIV TRANSMISSION (%) | | | | | |
| MSM[3] | 8,767 (40.1) | 6,224 (37.5) | 2,151 (45.2) | 66 (71.7) | 326 (77.8) |
| HET[4] | 7,152 (32.7) | 5,404 (32.5) | 1,654 (34.8) | 19 (20.7) | 75 (17.9) |
| I.V. drug use | 3,341 (15.3) | 2,834 (17.1) | 500 (10.5) | 0 (0.0) | 7 (1.7) |
| I.V. drug use/ sexual | 1522 (7.0) | 1288 (7.8) | 229 (4.8) | 2 (2.2) | 3 (0.7) |
| Perinatal transmission | 138 (0.6) | 100 (0.6) | 38 (0.8) | 0 (0.0) | 0 (0.0) |
| Other | 956 (4.4) | 756 (4.6) | 187 (3.9) | 5 (5.4) | 8 (1.9) |
| ETHNICITY (%) | | | | | |
| White | 14,818 (67.7) | 10,732 (64.6) | 3,650 (76.7) | 78 (84.8) | 358 (85.4) |
| Black | 2,414 (11.0) | 1,773 (10.7) | 620 (13.0) | 3 (3.3) | 18 (4.3) |
| Hispano-American | 682 (3.1) | 507 (3.1) | 143 (3.0) | 1 (1.1) | 31 (7.4) |
| Asian | 643 (2.9) | 448 (2.7) | 185 (3.9) | 4 (4.3) | 6 (1.4) |
| Other | 3,319 (15.2) | 3,146 (18.9) | 161 (3.4) | 6 (6.5) | 6 (1.4) |
| HIV SUBTYPE (%) | | | | | |
| B | 11,210 (51.2) | 7,815 (47.1) | 3,067 (64.4) | 27 (29.3) | 301 (71.8) |
| 02_AG | 717 (3.3) | 526 (3.2) | 182 (3.8) | 0 (0.0) | 9 (2.1) |
| C | 693 (3.2) | 512 (3.1) | 169 (3.6) | 1 (1.1) | 11 (2.6) |
| 01_AE | 652 (3.0) | 417 (2.5) | 195 (4.1) | 9 (9.8) | 31 (7.4) |
| A | 548 (2.5) | 496 (3.0) | 43 (0.9) | 4 (4.3) | 5 (1.2) |
| A1 | 349 (1.6) | 113 (0.7) | 208 (4.4) | 1 (1.1) | 27 (6.4) |
| G | 266 (1.2) | 194 (1.2) | 62 (1.3) | 3 (3.3) | 7 (1.7) |
| F | 140 (0.6) | 125 (0.8) | 9 (0.2) | 4 (4.3) | 2 (0.5) |
| D | 138 (0.6) | 101 (0.6) | 33 (0.7) | 0 (0.0) | 4 (1.0) |
| Other | 7,163 (32.7) | 6307 (38.0) | 791 (16.6) | 43 (46.7) | 22 (5.3) |
| PLASMA NGS SEQUENCE (%) | 2,161 (9.9) | 0 (0.0) | 1,755 (36.9) | 0 (0.0) | 406 (96.9) |
| PROVIRAL NGS SEQUENCE (%) | 3,184 (14.6) | 0 (0.0) | 3,064 (64.4) | 0 (0.0) | 120 (28.6) |

[1] Swiss HIV Cohort Study; [2] Zurich Primary HIV Infection Cohort Study; [3] Men Who Have Sex with Men; [4] Heterosexual.

[39]. In a randomized controlled trial on dolutegravir monotherapy in early treated PWH, the absence of proviral evolution and DRMs was confirmed [53,84]. It was further used for studying HIV-superinfection [43]. Further, the value of proviral diversity as a proxy for time since HIV infection was confirmed [81]. A viral genome wide association study (GWAS) and heritability analysis was performed which determined associations with neurocognitive complaints [82]. Multiple other projects are in progress, in particular attempts to infer the impact of the HIV genome on neurocognitive outcomes, immune responses, low-level viremia, to better identify hypermutations, and for proviral drug resistance testing.

## Discussion

Overall, established in early 2018, the SHCND has streamlined and standardized the viral genomic analyses in the Swiss HIV Cohort Study and the Zurich Primary HIV Infection

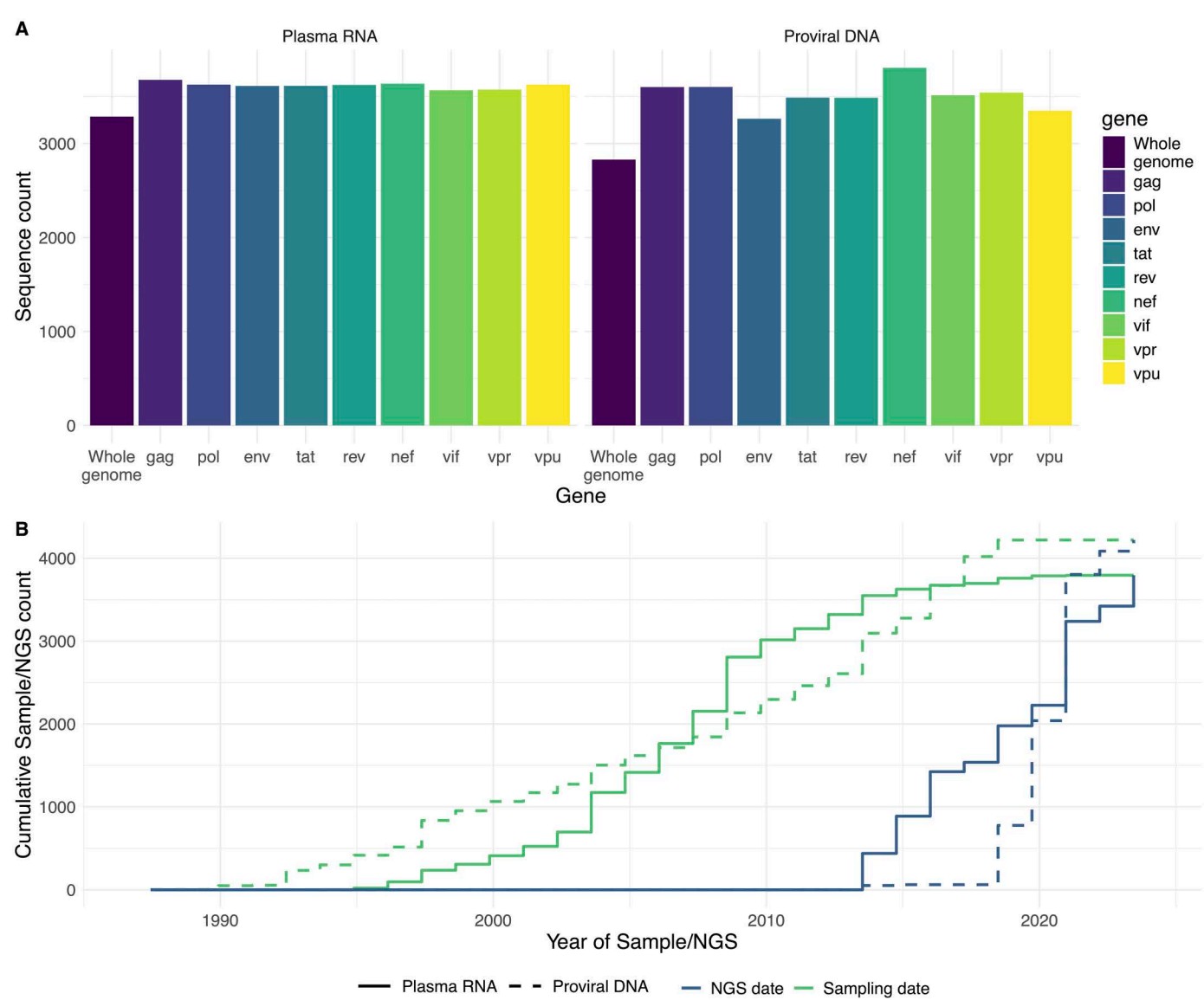

**Fig 4. Sequence availability and timing.** (A) Sequences available per gene and for the whole genome in the Swiss HIV Cohort Study Viral NGS Database (SHCND), stratified by sample source. Viral gene sequences considered, cover at least 40% of the respective gene length compared to the HIV-1 HXB2 gene reference sequence (GenBank accession number K03455). (B) Cumulative sample counts by sampling year and year when NGS was performed, stratified by HIV-1 plasma RNA and HIV-1 proviral DNA.

Cohort Study. Over its lifetime, the SHCND amassed 8,015 HIV-1 sequences from 5,178 PWH and continues to grow.

The SHCND is actively incorporating new bioinformatic tools based on the requests and requirements of researchers. Currently, most sequences in the database were obtained by Illumina MiSeq. The pipelines currently in use are designed for Illumina data. If it becomes a necessity to process sequences from other NGS technologies, the set-up of the database allows a quick implementation of other pipelines. Moreover, tools are maintained and improved to increase computational efficiency and usability. As tools dependent on the output of other tools must be executed in sequence, a current focus is the implementation of a piping functionality to allow the automatic execution of complex workflows. Recently, the SHCND has

**Table 3. Published original research which made use of the SHCND.**

| First author | Year | Journal | Title | Conclusion | Sequences used | Sample origin |
|---|---|---|---|---|---|---|
| Chaudron et al. [43] | 2022 | The Journal of Infectious Diseases | A Systematic Molecular Epidemiology Screen Reveals Numerous Human Immunodeficiency Virus (HIV) Type 1 Superinfections in the Swiss HIV Cohort Study | Prevalence of superinfections: 1%-7% | 128 | Plasma RNA |
| Balakrishna et al. [39] | 2023 | The Journal of Infectious Diseases | Frequency matters: comparison of drug resistance mutation detection by Sanger and next-generation sequencing in HIV-1 | High concordance of sanger sequencing and NGS for detection of HIV drug resistance mutations | 594 | Plasma RNA |
| Jörimann et al. [53] | 2023 | The Journal of Infectious Diseases | Absence of Proviral Human Immunodeficiency Virus (HIV) Type 1 Evolution in Early-Treated Individuals With HIV Switching to Dolutegravir Monotherapy During 48 Weeks | No HIV proviral evolution under dolutegravir monotherapy | 210 | Proviral DNA |
| Zeeb et al. [81] | 2024 | The Journal of Infectious Diseases | Genetic diversity from proviral DNA as a proxy for time since HIV-1 infection | HIV diversity of proviral DNA is a proxy for time since infection | 247 | Proviral DNA |
| Zeeb et al. [82] | 2024 | Brain Communications | Self-reported neurocognitive complaints in the Swiss HIV Cohort Study - a viral genome wide association study | Neurocognitive complaints in people with HIV are a heritable trait by the HIV genome | 2,334 | Plasma RNA and proviral DNA |

also gained the capability to run tools that analyze data from multiple NGS records in a single analysis, such as building phylogenetic trees, performing descriptive analyses on the whole sample archive of the SHCND or running BLAST searches. This type of processing can easily be split into a parallel preparation step that runs on individual samples and a final combination step.

We believe that the trend towards containerization of applications is very beneficial to avoid redundant efforts concerning software setup, installation, and dependency management. Therefore, we encourage the providers of such tools to include a containerized unit, as possible with Docker or Singularity, with their published data analysis tool. This has the advantage, that the analysis is done in a very reproducible manner, requiring only a Linux kernel, regardless of the state and configuration of the host operating system and installed programs outside the container.

We continuously evaluate the needs and costs of the system, to steward resources for updates. For instance, hardware requirements such as data storage and computational resources can fluctuate. Moreover, the underlying software which runs the SHCND, i.e., data management solutions, programming languages and libraries, high-performance computing clusters, or workflow managers have their own lifecycle and development that need to be handled. Crucially, these changes must keep the documentation history intact and are ideally done without users noticing it. This flexibility and customizability of the system is crucial for adapting to technological advancements. Moreover, as the core architecture is very general and not specifically tailored to NGS data and bioinformatic workflows, it allows for possible future expansions to include other data types. As such, we believe that our database solution would be beneficial in many other research settings where complex data is handled.

The implementation of the SHCND may in the future also incorporate other virus or pathogen genomes, for example, if there is an increase of coinfections, e.g., HCV or Mpox. There is also the potential to extend the SHCND even with human genome SNP data or antibody sequence data. Currently, our database already includes few Hepatitis C virus (HCV) genomes and the flexible database design would support seamless expansion. Furthermore, a separate instance of the database is currently being set-up for bacterial genomic data and corresponding analysis pipelines tailored to that, with the goal of conducting real-time molecular

epidemiolocal analyses to optimize infection control and facilitating genomic comparisons to better understand resistance spread mechanisms and support targeted interventions.

As human diseases are caused by complex interactions between host factors, environmental exposures, and microorganisms, research increasingly incorporates multiple data types, which require highly specialized computational tools, for example, multi-omics or one health approaches. We are particularly interested in metagenomic approaches, which would allow us to study the whole human microbiome including all infectious pathogens. We currently evaluate approaches in this direction, e.g., Kraken 2 [85]. Moreover, to investigate HIV transmission clusters we plan to integrate Phyloscanner [44]. Finally, while the current database has been established as a tool for research projects, it could also be readily adapted for clinical, diagnostic, and surveillance purposes in the future.

In conclusion, the SHCND considerably improved the scientific work with HIV NGS records in Switzerland. It reduced redundant computational processes and increased the reproducibility and accessibility of bioinformatic analyses. We hope the here outlined requirements, design, set-up, and use of a viral NGS database serves as inspiration and support for others with similar challenges.

## Author contributions

**Conceptualization:** Marius Zeeb, Paul Frischknecht, Huldrych F Günthard, Karin J Metzner, Roger D Kouyos.

**Data curation:** Paul Frischknecht, Suraj Balakrishna, Lisa Jörimann, Jasmin Tschumi, Sandra E Chaudron, Bashkim Jaha, Kathrin Neumann, Christine Leemann, Michael Huber, Karoline Leuzinger, Huldrych F Günthard, Karin J Metzner, Roger D Kouyos.

**Formal analysis:** Marius Zeeb, Paul Frischknecht.

**Funding acquisition:** Huldrych F Günthard, Karin J Metzner, Roger D Kouyos.

**Investigation:** Marius Zeeb, Paul Frischknecht, Huldrych F Günthard, Karin J Metzner, Roger D Kouyos.

**Methodology:** Marius Zeeb, Paul Frischknecht, Suraj Balakrishna, Lisa Jörimann, Jasmin Tschumi, Sandra E Chaudron, Bashkim Jaha, Huldrych F Günthard, Karin J Metzner, Roger D Kouyos.

**Project administration:** Huldrych F Günthard, Karin J Metzner, Roger D Kouyos.

**Resources:** Paul Frischknecht, Suraj Balakrishna, Huldrych F Günthard, Karin J Metzner, Roger D Kouyos.

**Software:** Paul Frischknecht.

**Supervision:** Huldrych F Günthard, Karin J Metzner, Roger D Kouyos.

**Validation:** Marius Zeeb, Paul Frischknecht, Lisa Jörimann, Jasmin Tschumi, Levente Zsichla, Sandra E Chaudron, Bashkim Jaha, Kathrin Neumann, Christine Leemann, Michael Huber, Karoline Leuzinger, Huldrych F Günthard, Karin J Metzner, Roger D Kouyos.

**Visualization:** Marius Zeeb.

**Writing – original draft:** Marius Zeeb, Paul Frischknecht, Huldrych F Günthard, Karin J Metzner, Roger D Kouyos.

**Writing – review & editing:** Marius Zeeb, Paul Frischknecht, Suraj Balakrishna, Lisa Jörimann, Jasmin Tschumi, Levente Zsichla, Sandra E Chaudron, Bashkim Jaha, Kathrin Neumann, Christine Leemann, Michael Huber, Karoline Leuzinger, Huldrych F Günthard, Karin J Metzner, Roger D Kouyos.

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
