## [Decision Letter · Decision Letter 0]

3 Dec 2024

PDIG-D-24-00350Addressing data management and analysis challenges in viral genomics: The Swiss HIV Cohort Study Viral Next Generation Sequencing databasePLOS Digital Health Dear Dr. Zeeb, Thank you for submitting your manuscript to PLOS Digital Health. After careful consideration, we feel that it has merit but does not fully meet PLOS Digital Health's publication criteria as it currently stands. Therefore, we invite you to submit a revised version of the manuscript that addresses the points raised during the review process. Please submit your revised manuscript within 60 days Feb 01 2025 11:59PM. If you will need more time than this to complete your revisions, please reply to this message or contact the journal office at digitalhealth@plos.org. Please include the following items when submitting your revised manuscript:* A rebuttal letter that responds to each point raised by the editor and reviewer(s). You should upload this letter as a separate file labeled 'Response to Reviewers '. This file does not need to include responses to any formatting updates and technical items listed in the 'Journal Requirements' section below.* A marked-up copy of your manuscript that highlights changes made to the original version. You should upload this as a separate file labeled 'Revised Manuscript with Track Changes '.* An unmarked version of your revised paper without tracked changes. You should upload this as a separate file labeled 'Manuscript '. If you would like to make changes to your financial disclosure, competing interests statement, or data availability statement, please make these updates within the submission form at the time of resubmission. Guidelines for resubmitting your figure files are available below the reviewer comments at the end of this letter. We look forward to receiving your revised manuscript. Kind regards, Miguel Ángel Armengol de la Hoz, Ph.D.Section EditorPLOS Digital Health Leo Anthony CeliEditor-in-ChiefPLOS Digital Healthorcid.org/0000-0001-6712-6626 **Journal Requirements:**

**Please only choose the relevant sentences from below**

1. Please clarify all sources of funding (financial or material support) for your study. List the grants (with grant number) or organizations (with url) that supported your study, including funding received from your institution. 

2. State the initials, alongside each funding source, of each author to receive each grant.

3. State what role the funders took in the study. If the funders had no role in your study, please state: “The funders had no role in study design, data collection and analysis, decision to publish, or preparation of the manuscript.”

4. If any authors received a salary from any of your funders, please state which authors and which funders.

2. Please send a completed 'Competing Interests' statement, including any COIs declared by your co-authors. If you have no competing interests to declare, please state "The authors have declared that no competing interests exist". Otherwise please declare all competing interests beginning with the statement "I have read the journal's policy and the authors of this manuscript have the following competing interests:"

3. Please note that your Data Availability Statement is currently missing the repository name and the DOI/accession number of each dataset OR a direct link to access each database. If your manuscript is accepted for publication, you will be asked to provide these details on a very short timeline. We therefore suggest that you provide this information now, though we will not hold up the peer review process if you are unable.

4. We ask that a manuscript source file is provided at Revision. Please upload your manuscript file as a .doc, .docx, .rtf or .tex.

5. Please provide an Author Summary. This should appear in your manuscript between the Abstract (if applicable) and the Introduction, and should be 150–200 words long. The aim should be to make your findings accessible to a wide audience that includes both scientists and non-scientists. Sample summaries can be found on our website under Submission Guidelines: 

https://journals.plos.org/digitalhealth/s/submission-guidelines#loc-parts-of-a-submission

**Additional Editor Comments (if provided):****Reviewers' Comments:** Reviewer's Responses to Questions

**Comments to the Author**

1. Does this manuscript meet PLOS Digital Health’s publication criteria ? Is the manuscript technically sound, and do the data support the conclusions? The manuscript must describe methodologically and ethically rigorous research with conclusions that are appropriately drawn based on the data presented.

Reviewer #1: Partly

Reviewer #2: Yes

Reviewer #3: Yes

2. Has the statistical analysis been performed appropriately and rigorously?

Reviewer #1: N/A

Reviewer #2: I don't know

Reviewer #3: N/A

3. Have the authors made all data underlying the findings in their manuscript fully available (please refer to the Data Availability Statement at the start of the manuscript PDF file)?

Reviewer #1: Yes

Reviewer #2: Yes

Reviewer #3: Yes

4. Is the manuscript presented in an intelligible fashion and written in standard English?

Reviewer #1: Yes

Reviewer #2: Yes

Reviewer #3: Yes

5. Review Comments to the Author

Reviewer #1: The manuscript by Zeev et al. offers an in-depth exploration of how HIV NGS data are processed and managed within the Swiss HIV Cohort Study. The authors have previously contributed several notable studies using NGS data, as highlighted in Table 2.

While certain sections provide an excess of detail unlikely to be of broad interest, other critical topics receive insufficient attention. Furthermore, the specific aims of the manuscript are not clearly articulated.

1. Data analysis and amplification methodology: The approach to data analysis and management depends significantly on the method by which the virus is amplified. Historically, sequencing of the HIV pol gene has relied on sets of well-conserved primers. However, achieving full-genome sequencing, especially of highly variable regions, presents additional challenges. The manuscript does not discuss how the analysis pipeline accounts for the amplification method used to generate whole-genome data. This omission raises doubts about the assertion in the abstract that "NGS can easily achieve near-whole-genome sequence coverage." In fact, several studies reporting full-genome sequencing have employed different strategies, underscoring the complexity involved.

2. NGS platform variability: There is no discussion of how the choice of NGS platform influences the analysis. Pipelines optimised for Illumina, PacBio, or ONT platforms can differ substantially, and this is not addressed in the manuscript.

3. Use of unique molecular identifiers (UMIs): While some research groups employ UMIs to reduce error and bias, it seems the described pipeline is not designed to accommodate sequences tagged with UMIs. This limitation warrants mention. Without UMIs, the claim that NGS "accurately encapsulates within-host diversity by characterising HIV subpopulations" is likely overstated.

4. Clinical versus research applications: It is unclear whether the pipeline described is employed for clinical drug resistance testing, and if so, how it differs from its use in research. Clarifying this distinction would enhance the manuscript.

5. Quality control challenges: Quality control is only briefly touched upon, with a passing reference to the Hypermut programme. This is particularly relevant for sequences derived from proviral DNA, where maintaining sequence integrity poses significant challenges.

In summary, while the manuscript offers valuable insights, it would benefit from a clearer focus on the practical relevance of the described methods, with additional attention to the amplification methods, NGS platforms, and quality control measures essential for both clinical and research contexts.

Reviewer #2: The study presents the innovative solution of SHCND in a sequential easy to follow fashion. I find the methods used detailed in a well streamed flow. The results presented a promising solution to address challenges in the HIV genome data management. Especially in standardizing the steps in handling the data, processing them and retrieving them for research or follow up purposes. The solution presented deserves population to set an example for wider global implementation and further modification to fit the diversity of the features of HIV infection, and might even be applicable for other common infectious diseases world wide.

In this context, I would suggest adding the prospects of widely sharing the achieved goals of the solution and the opportunities of collaboration to widely develop and implement the database solution.

One modification to be suggested is the presentation of the references in the introduction, methods and results sections. The citation within the text is distracting, and I would suggest that citation with putting the number of the cited reference in the text. Especially when citing multiple references.

The study also mentioned that some sample categories were over-represented without giving precise details about which categories are overrepresented and upon the expense of which underrepresented categories.

The references that presented the preparation, amplification, and sequencing protocols were cited. However, the matching and comparison of these protocols in this study and in the cited references were not detailed.

Finally the language used could be made easier for lay reader, which is something that I always advocate in reviewing any publication, because I believe it will not only benefit reader, but also benefit the published study in giving it wider outreach

Reviewer #3: Please see comments/suggestions

Introduction

1. The revised version should use PLOS Digital Health's citation style.

2. Paragraph 2 should be broken down into 2-3 paragraphs. The general public may have a difficult time digesting this.

3. Paragraph 2 shows some obstacles in setting up HIV-related databases but does not provide the specific databases on where these obstacles come from. Are there existing databases (whether HIV-related or not) that you can cite to justify why your database is superior?

4. I feel that part of the intro should include the extent by which the development and deployment of the NGS database adheres to FAIR Guidelines: https://pmc.ncbi.nlm.nih.gov/articles/PMC4792175/

Methods

5. Most of the arrowheads in Figure 2 are so small that it is difficult to determine their direction. Moreover, just use the full term for BioInf unless you plan to place a legends section.

6. Please provide a citation for the advantages of using UUIDv4

7. Figure 1 does not show this process: "All generated downstream data files receive their own UUID as well

and are linked to the corresponding NGS record and processing tool through additional metadata files." I also feel that Figure 1 needs to be expanded to provide a complete description of the workflow involved in using the database.

8. The manuscript will benefit from the addition of additional figures that visualize the contents of each methods sub-section. For now, only the workflow has a visual (which in itself seems to be lacking too).

9. Is the data availability statement found online?

Discussion

10. The results do not reflect the statement in paragraph 1 of the discussion section. Can you provide evidence that it's use makes the workflow efficient? While the technical details are presented in the methods section, using the term efficient is farfetched since no information was presented before and after.

6. PLOS authors have the option to publish the peer review history of their article (what does this mean? ). If published, this will include your full peer review and any attached files.

**Do you want your identity to be public for this peer review?** For information about this choice, including consent withdrawal, please see our Privacy Policy .

Reviewer #1: No

Reviewer #2: **Yes: ** Yasser Abdullah

Reviewer #3: No

---

## [Decision Letter · Decision Letter 1]

16 Mar 2025

Addressing data management and analysis challenges in viral genomics: The Swiss HIV Cohort Study Viral Next Generation Sequencing database

PDIG-D-24-00350R1

Dear Mr. Zeeb,

We are pleased to inform you that your manuscript 'Addressing data management and analysis challenges in viral genomics: The Swiss HIV Cohort Study Viral Next Generation Sequencing database' has been provisionally accepted for publication in PLOS Digital Health.

Best regards,

Miguel Ángel Armengol de la Hoz, Ph.D.

Section Editor

PLOS Digital Health

**Additional Editor Comments (if provided):**

**Reviewer Comments (if any, and for reference):**

Reviewer's Responses to Questions

**Comments to the Author**

1. If the authors have adequately addressed your comments raised in a previous round of review and you feel that this manuscript is now acceptable for publication, you may indicate that here to bypass the “Comments to the Author” section, enter your conflict of interest statement in the “Confidential to Editor” section, and submit your "Accept" recommendation.

Reviewer #2: All comments have been addressed

Reviewer #3: All comments have been addressed

2. Does this manuscript meet PLOS Digital Health’s publication criteria ? Is the manuscript technically sound, and do the data support the conclusions? The manuscript must describe methodologically and ethically rigorous research with conclusions that are appropriately drawn based on the data presented.

Reviewer #2: Yes

Reviewer #3: Yes

3. Has the statistical analysis been performed appropriately and rigorously?

Reviewer #2: Yes

Reviewer #3: N/A

4. Have the authors made all data underlying the findings in their manuscript fully available (please refer to the Data Availability Statement at the start of the manuscript PDF file)?

Reviewer #2: Yes

Reviewer #3: Yes

5. Is the manuscript presented in an intelligible fashion and written in standard English?

Reviewer #2: Yes

Reviewer #3: Yes

6. Review Comments to the Author

Reviewer #2: Thanks for addressing all the review suggestions

Reviewer #3: I am satisfied with the revision. Thank you.

7. PLOS authors have the option to publish the peer review history of their article (what does this mean? ). If published, this will include your full peer review and any attached files.

**Do you want your identity to be public for this peer review?** For information about this choice, including consent withdrawal, please see our Privacy Policy .

Reviewer #2: **Yes: ** Yasser Abdullah

Reviewer #3: No
